# A Qualitative Dataset for Coffee Bio-Aggressors Detection Based on the Ancestral Knowledge of the Cauca Coffee Farmers in Colombia

Juan Felipe Valencia-Mosquera [1,*], David Griol [2,*], Mayra Solarte-Montoya [3], Cristhian Figueroa [1], Juan Carlos Corrales [1] and David Camilo Corrales [4,*]

1    Departamento de Telemática, Universidad del Cauca, Popayán 190002, Colombia; cfigmart@unicauca.edu.co (C.F.); jcorral@unicauca.edu.co (J.C.C.)
2    Department of Software Engineering, University of Granada, 18010 Granada, Spain
3    Grupo de Investigaciones para el Desarrollo Rural TULL, Facultad de Ciencias Agrarias, Universidad del Cauca, Popayán 190002, Colombia; mayrasolarte@unicauca.edu.co
4    French National Research Institute for Agriculture, Food and Environment—INRAE, UMS (1337) TWB, 135 Avenue de Rangueil, 31077 Toulouse, France; david-camilo.corrales-munoz@inrae.fr
*    Correspondence: juanvalencia@unicauca.edu.co (J.F.V.-M.); dgrioll@ugr.es (D.G.)

**Abstract:** This paper describes a novel qualitative dataset regarding coffee pests based on the ancestral knowledge of coffee farmers in the Department of Cauca, Colombia. The dataset has been obtained from a survey applied to coffee growers with 432 records and 41 variables collected weekly from September 2020 to August 2021. The qualitative dataset includes climatic conditions, productive activities, external conditions, and coffee bio-aggressors. This dataset allows researchers to find patterns for coffee crop protection through the ancestral knowledge not detected by real-time agricultural sensors. As far as we are concerned, there are no datasets like the one presented in this paper with similar characteristics of qualitative value that express the empirical knowledge of coffee farmers used to detect triggers of causal behaviors of pests and diseases in coffee crops.

**Dataset:** https://doi.org/10.5281/zenodo.8275090.

**Dataset License:** Licensed under Creative Commons Attribution 4.0 International (CC-BY-4.0).

**Keywords:** ancestral knowledge; coffee crops; coffee pest; coffee diseases; coffee bio-aggressors; weather conditions; qualitative crop data



## 1. Summary

The Department of Cauca in Colombia presents ideal environmental conditions for *Coffea arabica* L. cultivation known for its light and flowery taste. In this region, about 43,000 Afro-descendant, indigenous, and peasant families planted almost 44,500 hectares of arabica varieties: Castillo, Colombia, Caturra, Típica, Borbón, and Tabí [1–4].

Currently, coffee farming faces many challenges related to climate change, such as a higher incidence of pests and plant pathogens, leading to lower productivity [5]. The most damaging coffee pests are coffee rust (*Hemileia vastatrix*), Coffee Berry Borer (CBB) (*Hypothenemus hampei*), and coffee spot (*Cercospora coffeicola*). These pests cause lesions on leaves and fruits, reducing coffee yields [6,7].

Although most coffee-cultivated regions in the department of Cauca are located above 1500 m, making them less vulnerable to pests and diseases; about 8.4% are located below 1400 m (approx. 3200 ha), which are still vulnerable [8]. Coffee rust affects 28.5% of these regions with susceptible varieties [7,9], while coffee berry borer and brown-eye spots affect 5.0% and 4.8%, respectively, of the coffee crops [10]. Furthermore, climatic

conditions, cultivation practices, and the local landscape contribute to the outbreaks of these pests [11–13].

In this context, monitoring tools are essential to ease crop protection, which has been addressed with intelligent machines and sensors that generate large amounts of real-time agricultural data (e.g., soil, weather, pests, and plants [14–18]). Nonetheless, coffee-growing families in the Cauca department are accustomed to making decisions about protecting their coffee crops by trusting ancestral knowledge rather than data from sensors and intelligent machines [19–21].

Ancestral knowledge (also known as peasant knowledge) refers to the knowledge acquired empirically over the production of crops, transmitted from generation to generation of farmers in order to face problems that affect their crops [22,23]. In Latin America, different Afro-descendant, indigenous, and peasant families have developed popular wisdom by observing the components of their environment. Therefore, ancestral knowledge has been acquired through observation and experimentation based on various crops' nutritional, medicinal, or confectionery usefulness.

This paper presents a qualitative approach to gathering ancestral knowledge regarding bio-aggressor control in coffee cultivation within the Cauca department. The dataset is derived from a semi-structured survey conducted among coffee farmers from the Association of Agricultural Producers of Cajibio (ASPROACA). The dataset extensively captures qualitative practices, including crop management, labor strategies, bioindicator utilization, and traditional wisdom. Additionally, it encompasses influential external factors, such as climate conditions, which play a pivotal role in crop outcomes. This comprehensive dataset facilitates the correlation of variables and attributes with pest and disease occurrences, enabling experts and farmers to gain deeper insights into pest management dynamics.

The paper continues as follows: Section 2 outlines the methodology for data collection, including details on the study area, survey questions, and participant recruitment. Section 3 highlights the variables and metadata encompassed within the dataset. Moving to Section 4, we delve into data processing and weighting the survey's data. Lastly, Section 5 encapsulates the key conclusions and outlines prospects for future research works.

*Background*

- Coffee Crop phenology: Crop phenology defines the physiological development stages of crop growth from planting to harvest [24]. Colombian farmers rely on crop phenology, which includes the coffee flowering stage, to make decisions related to agronomic practices such as fertilization, water deficit, pest, and disease management [25]. The coffee flowering stage also plays a crucial role in determining the number of harvests and their distribution throughout the year. Furthermore, flowering allows the identification of the critical periods of CBB infestations, water irrigation periods, and nutrients for the crop's needs. Due to climatic conditions and relief in Colombia, there are different flower and fruit development periods during the year.

Table 1 shows the beginning of the flowering and harvest during the year. In southwest Colombia, there are two flowering seasons; therefore, there are two harvests (the first harvest of the year, named in Spanish 'mitaca', and the main harvest). In the mitaca, the flowering begins in September and finishes in November. After the first flowering, the growth and maturation of the fruit are presented between March and May, corresponding to the year's first harvest. The second flowering begins between February and April, followed by the main harvest between September and November.

Due to coffee being a perennial crop [24], it is considered a complete crop cycle between the flowering and post-harvest stages. The information in the dataset presented in this paper was collected during the coffee growing cycle, including the flowering, ripening, and harvesting until the post-harvesting stages of the fruit.

- Study area: The dataset presents data collected through a questionnaire survey by making phone calls. The survey was answered by participants of nine coffee farmers

from the association ASPROACA [26]. ASPROACA is located in Cajibio, as depicted in Figure 1.

**Table 1.** Main flowering and harvest seasons in Colombian coffee crops.

| Seasons of Flowering and Harvest in Colombia | 1st Flowering | 1st Harvest (Mitaca) | 2nd Flowering | 2nd Harvest |
|---|---|---|---|---|
| January | | | | |
| February | | | X | |
| March | | X | X | |
| April | | X | X | |
| May | | X | | |
| June | | | | |
| July | | | | |
| August | | | | |
| September | X | | | X |
| October | X | | | X |
| November | X | | | X |
| December | | | | |

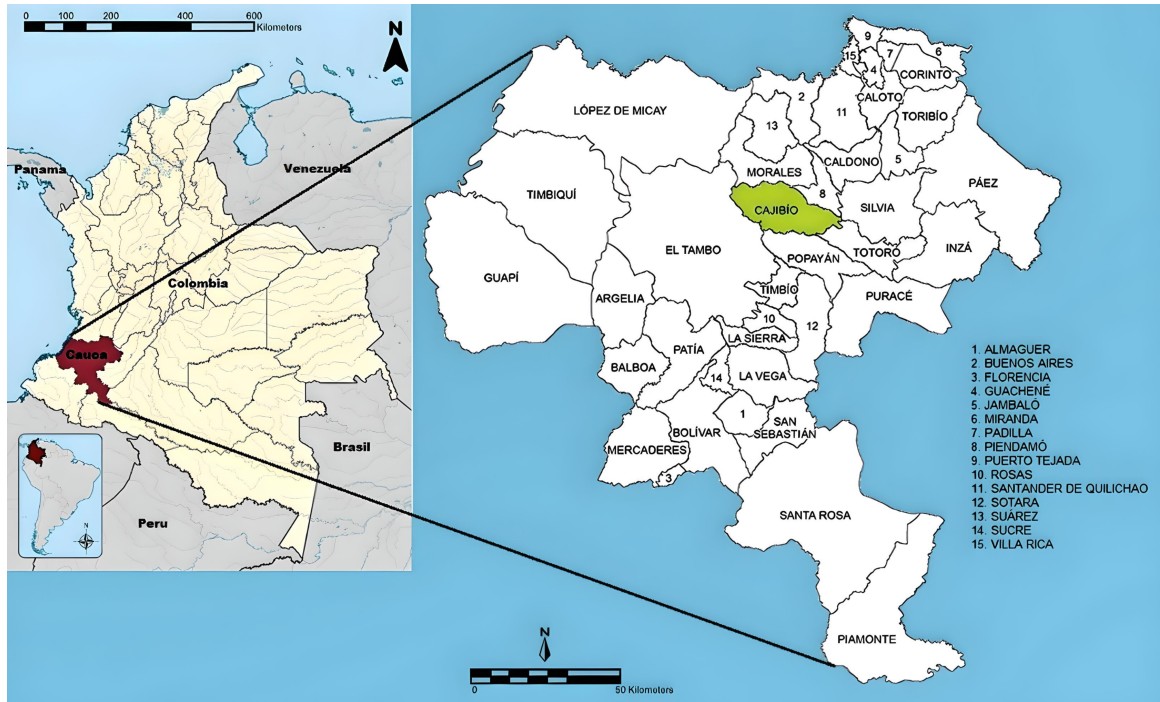

**Figure 1.** The left side highlights the Department of Cauca in red, while the right side showcases the location of Cajibio, indicated in green. The list of numbers on the image indicates the name in Spanish of each township located graphically within the department of Cauca, Colombia.

The collected data relied on coffee farmers' voluntary cooperation, providing information on their folk wisdom or ancestral knowledge about their productive work in coffee crops acquired from generation to generation. Table 2 presents the respondents' information and their coffee crops.

**Table 2.** Data summary: coffee farmers and their crops.

| Coffee Farmer ID | Gender | Age | Coffee Planted (ha) | Coffee Variety | Productive Arrangement |
|---|---|---|---|---|---|
| CF-01 | M | 52 | 2.5 | Caturra | Under free shadow exposure |
| CF-02 | F | 48 | 4.0 | Tabi | Under shadow conditions |
| CF-03 | M | 39 | 3 | Castillo | Under free sun exposure |
| CF-04 | F | 46 | 3.8 | Castillo | Under free shadow exposure |
| CF-05 | M | 36 | 2 | Caturra | Under free shadow exposure |
| CF-06 | F | 54 | 4 | Tabi | Under free shadow exposure |
| CF-07 | F | 72 | 1 | Caturra | Under free shadow exposure |
| CF-08 | M | 36 | 7 | Caturra | Under free sun exposure |
| CF-09 | M | 50 | 1 | Tabi | Under free shadow exposure |

## 2. Data Description

The information recollected through the survey format was transformed into plain text separated by comma values (CSV file). The dataset comprises qualitative/numeric variables such as climatic conditions, the productive activities of the coffee crops, ancestral knowledge, and other external conditions found in the crops (Table 3).

**Table 3.** Quantitative variables of the dataset.

| Variable | Description | Values | Quantity |
|---|---|---|---|
| 1. Coffee grower | Coffee grower ID | Number of coffee grower | 9 |
| 2. Age | Age of coffee grower | Integer value between 36 and 72 | (35–49) = 5, (50–64) = 3, (≥65) = 1 |
| 3. Gender | Gender of coffee grower | Male/Female | Male = 5, Female = 4 |
| 4. Crop hectares | Hectares of cultivation | Decimal value between 1.0 and 7.0 hectares | [1, 3) = 192, [3, 5) = 192, [5, 7] = 48 |
| 5. Survey date | Date of survey application | Day/Month/Year—DD/MM/YYYY | |
| 6. Survey month | Month of survey application | Month of the year | |
| 7. Survey week | Week of the survey application | First, Second, Third, Fourth | |
| 8. Production arrangement | Crop distribution | Under free shadow exposure, Under shadow conditions, Under free sun exposure | 48, 288, 96 |
| 9. Crop phenology | Crop phenological stage | Flowering, fruit ripening, harvest, post-harvest | 144, 144, 72, 72 |
| 10. Soil level | Slope of the farmland | Contour planting, flat cultivation | 288, 144 |
| 11. Crop variety | Variety of the crop used | In Spanish (Castillo, Caturra, Tabi) | 96, 192, 144 |
| 12. Temperature | Temperature on crop | Cold, Heat | 172 , 196 |
| 13. Rainfall intensity | Intensity of rainfall on the crop | Hailstorm, Rains, Dry | 4, 192, 97 |
| 14. Crop nutrition | Crop nutrition | Yes/Not | No = 417 Yes = 15 |
| 15. Collection and review | Labor post-harvest done | Yes/Not | No = 253 Yes = 179 |
| 16. Zoca | Cutting the coffee tree | Yes/Not | No = 417 Yes = 15 |
| 17. Fungicides use | Proper use of fungicides | Yes/Not | No = 378 Yes = 54 |
| 18. Biopesticides use | Proper use of Biopesticides | Yes/Not | No = 423 Yes = 9 |
| 19. Crop rotation | Crop rotation | Yes/Not | No = 395 Yes = 37 |
| 20. Intercropping | Coffee inter-cropped with avocado, guamo, and others | Yes/No/Others | No = 341 Yes = 91 |
| 21. Weed control | Weed control in crops | Yes/Not | No = 297 Yes = 135 |

**Table 3.** *Cont.*

| Variable | Description | Values | Quantity |
|---|---|---|---|
| 22. Ancestral knowledge | Lunar stages and other productive labors on crop | Yes/No | No = 336 Yes = 96 |
| 23. External conditions | Nearby rivers or mammals and birds near to crop | Yes/Not | No = 288 Yes = 144 |
| 24. Pest | Pest Coffee | Coffee Berry Borer, berry blotch | 128, 4 |
| 25. Diseases | Diseases triggered in the crop | Brown-eye spot | 44 |
| 26. Weed | Crop weeds type | Aggressive weeds, Noble Weeds | 2, 254 |

The dataset comprises 432 records and 41 variables. Temperature and rainfall intensity are associated variables with weather conditions. Crop management, biopesticide control, polyculture, ancestral knowledge, crop phenology, Zoca, productive arrangement, and intercropping variables are related to productive activities. Variables like animals close to the crop, landscape influence, and water sources reflect external conditions. Lastly, the pests, diseases, and weeds variables compose bio-aggressors.

## 3. Methods

Different activities were carried out at this stage in order to develop the dataset proposed, starting with the detection and definition of the hypothesis: How can we detect qualitative information regarding bio-aggressor control in coffee crops based on ancestral knowledge? After that, a review was carried out of the existing scientific literature in order to establish a solid knowledge base, the selection definition and establishment of the research design, and population. Data collection was performed through semi-structured interviews and the consolidation of the logbook. The processing of the data was obtained through different approaches to discover behavior within the data. We carried out the interpretation and analysis of the results and, finally, drew conclusions. All of those tasks were included in the methodological approach shown on a large scale in Figure 2:

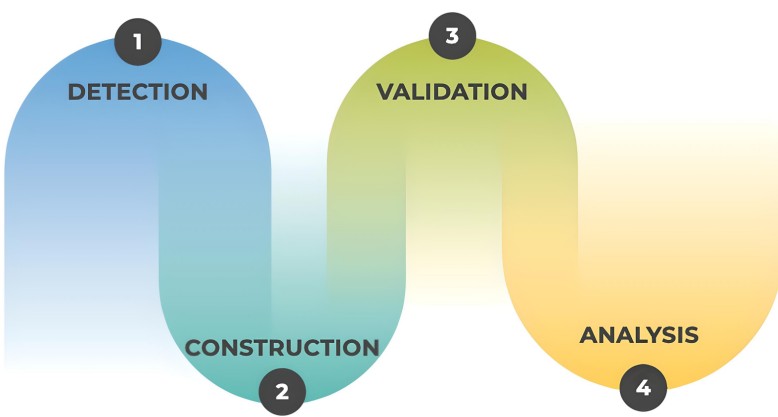

**Figure 2.** Methodological approach.

### 3.1. Detection

The detection process focused on identifying qualitative information related to productive activities and coffee bio-aggressors, intending to define the variables to be considered in the dataset. As the primary process, exhaustive bibliographic reviews of scientific documents between the years 1990 and 2022 were carried out, establishing inclusion criteria for the scientific articles and exclusion criteria for non-formal documents. Initially, 40 research studies were obtained; however, under the exclusion criteria, 27 papers were selected that directly related to productive tasks with bio-aggressors in coffee. The analysis of the results of the most representative scientific articles on bio-aggressors in coffee due to productive activities are summarized in Figure 3 below.

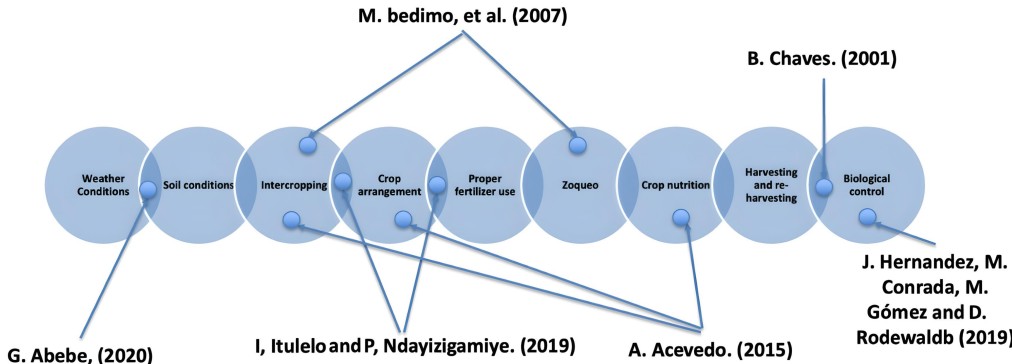

**Figure 3.** Set of significant variables generated from literature reviews [27–32].

*3.2. Construction*

Figure 4 illustrates the developmental process we pursued to create our dataset.

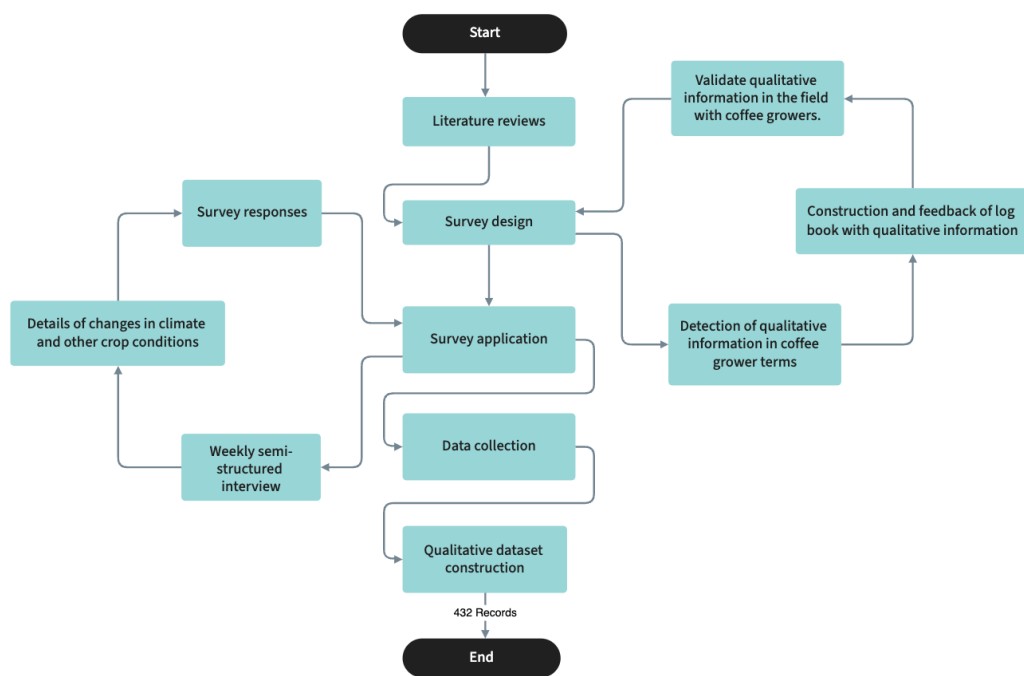

**Figure 4.** Dataset construction process.

The construction process consists of four main stages. After defining the basic variables to be taken into account during the analysis stage of the methodological approach, it starts with survey design, survey implementation, data collection, and, finally, the construction of the dataset.

The survey design began with activities such as filling out a logbook, which allowed the terms that coffee growers use about their crops to be stored. Filling out the log allowed us to establish a more user-friendly knowledge base with exploratory purposes to acquire a greater knowledge about coffee cultivation and identify how they visualized the different variables established from the bibliographic analysis. Different activities that coffee growers carried out periodically in their crops were identified. This valuable information was filled out in the record book, allowing it to be one of the main inputs to establish and design the semi-structured surveys with the support of the research group for rural development TULL [26], framing the components or variables of the climatic conditions, the productive tasks, external crop conditions, and bio-aggressors of coffee cultivation in Cauca.

Regarding the survey application with the process of applying the semi-formal interviews, it was decided to conduct them weekly in order to monitor the productive work

carried out continuously, the weather conditions on the crop, and the validation of other external variables. Mainly, the interviews were based on monitoring events such as weather conditions, and the respective productive work carried out that week. The main objective was to continuously survey possible climatic changes associated with other variables that would allow important information to be saved regarding coffee bio-aggressors. Each coffee farmer's interview enriched the dataset with all the information provided. The semi-formal interview process took into account a complete coffee phenological cycle of the Cauca coffee crop, as shown in Figure 5.

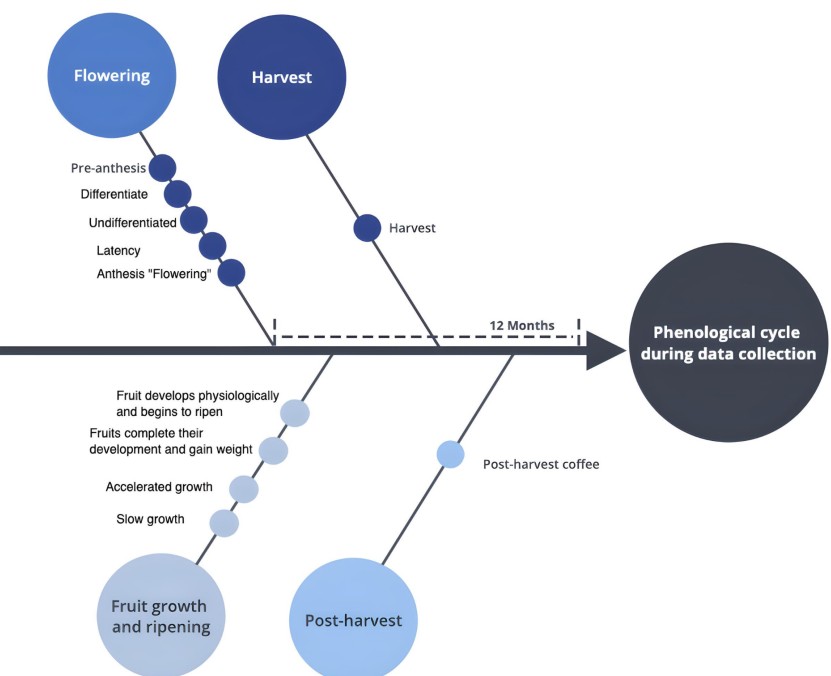

**Figure 5.** Phenological cycle during data collection.

As the final step of the dataset construction process, the data collection involved the active participation of nine coffee growers from the ASPROACA association. This process heavily relied on the voluntary collaboration of the farmers, who generously shared information passed down through generations. They provided insights into variables encompassing climate conditions, productive activities, external crop conditions, and bio-aggressors' presence within their crops.

### 3.3. Validation

Different activities were carried out in the validation process, including verifying whether the log book was meeting its objective of establishing a common language that was easily understood by the coffee grower, paving the way for the creation of the survey results, and leading to the study of the variables for the detection of causal bio-aggressors. Several iterations were carried out with the coffee growers, ensuring the correct terminology for the data generation from the surveys and the subsequent establishment of the qualitative dataset based on the information provided. The validation process generated constant feedback, facilitating the acquisition of valuable information appropriate for the qualitative dataset scheme constructed.

In summary, the validation consisted of a process of continuous feedback where, in direct dialogue with the coffee growers, different topics about the crop were discussed. All the information that the coffee growers provided increasingly strengthened the different uses of the empirical knowledge of the productive tasks carried out by the coffee growers.

Finally, constructing the dataset focuses on filling out all the information collected in the log book and under the semi-formal interviews. The centralization of information

resulted in the archiving of the dataset. As time passed, it was enriched with a more significant number of records.

### 3.4. Analysis

The analysis stage of the methodological approach allows us to examine, interpret, and synthesize to draw meaningful conclusions or insights. This phase is crucial for transforming raw data into actionable knowledge and generating valid and reliable conclusions.

Within the present phase, the following aspects were taken into account:

- Data preparation: Data was prepared by organizing and cleaning the data to set up the analysis under different data mining approaches and statistical criteria. Classification tasks, coding, and information structuring were performed in the data preparation to facilitate subsequent analysis.
- Data exploration: An exploration of the data obtained allowed for establishing a characterization of the variables detailed in Figure 6.

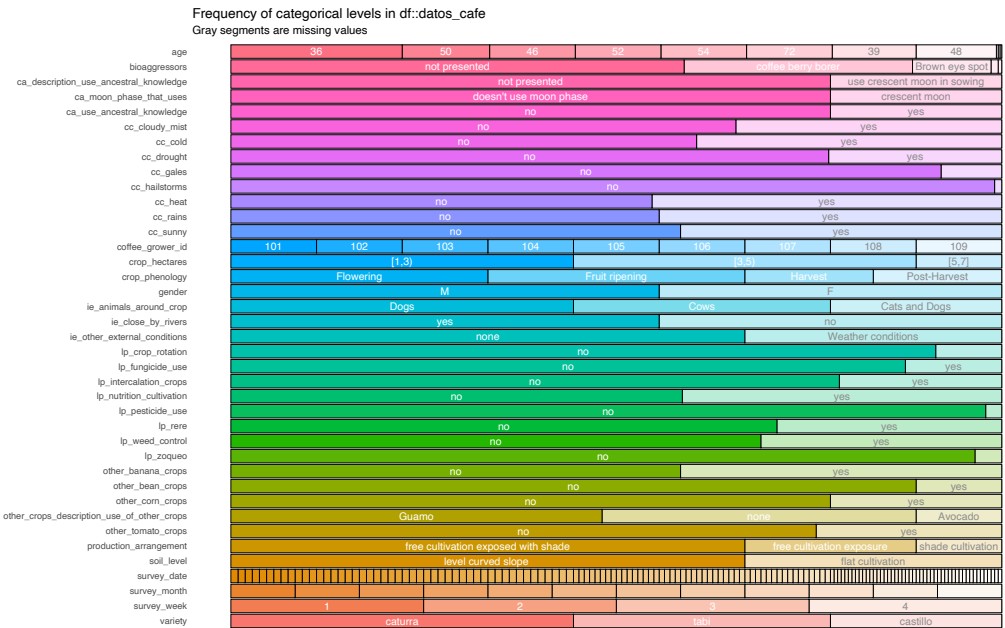

**Figure 6.** Characterization and frequencies of each variable in the dataset.

- Data transformation or pre-processing: The raw data were processed to make them suitable for analysis. This involved normalizing, standardizing, aggregating, or converting the data into a more interpretative form Figure 7.

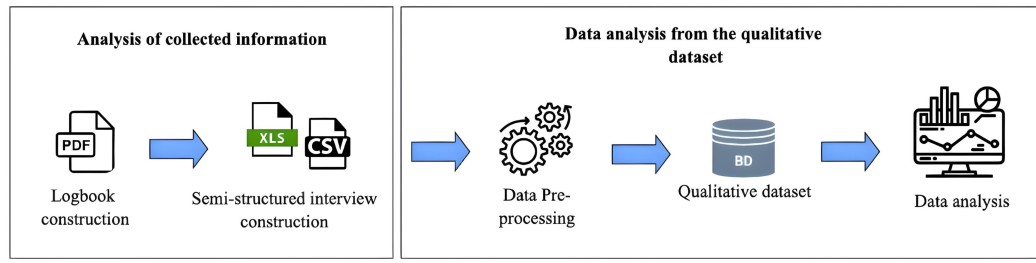

**Figure 7.** Data transformation.

- Statistical or analytical techniques: Understanding the qualitative value of the variables defined the analysis under descriptive statistical techniques such as multiple component analysis, data mining approaches such as hierarchical clustering under dendrograms, the application of the wrapper selection method, and the application of

decision trees under the application of C4.5 algorithm. The details of the applications are shown in more detail (Section 3.6, Data analysis), aiming to evaluate and obtain inferences or discoveries of behavioral patterns that occur within the data.

### 3.5. Survey Content

Surveys were scheduled by each farmer according to their availability and conducted weekly from September 2020 to August 2021, obtaining 432 records. This survey divides the main agronomic characteristics into categories and is displayed in Figure 8.

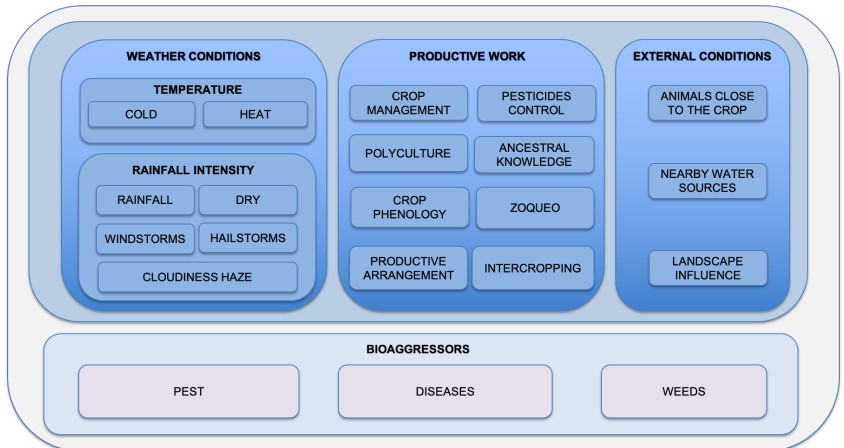

**Figure 8.** Variables and corresponding categories considered in the survey.

### 3.5.1. Weather Conditions

High temperatures and sudden precipitation influence the appearance of coffee bio-aggressors. If a coffee crop has high temperatures due to sunshine, but there is a sudden climatic change with precipitation, a humid environment is created, increasing the possibility of triggering bio-aggressors in the crop [33]. Therefore, climatic conditions play an important role in coffee, and Table 4 shows the climatic variables considered in the study. The column "Week of the month" indicates the week of a month in which the weather event occurred. The table was filled out for each month during data collection.

**Table 4.** Weather variables surveyed weekly.

| Weather Variable | Qualitative Value | Week of the Month |
|---|---|---|
| Temperature | Heat | Week 1 |
| | Cold | Week 2 |
| Rainfall intensity | Rainfall | |
| | Dry | Week 3 |
| | Windstorms | |
| | Hailstorms | |
| | Cloudiness/haze | Week 4 |
| Other | Other conditions | |

The survey has incorporated three qualitative values for the temperature variable: hot and cold. For rainfall intensity, the model encompasses five qualitative values: rainfall, dry, windstorms, hailstorms, and cloudiness/haze.

### 3.5.2. Productive Activities

Coffee growers engage in various productive activities related to crop management, pesticide control, polyculture, ancestral knowledge, crop phenology, zoca, productive arrangement, and intercropping. Refer to Table 5 for a detailed breakdown of these activities.

The column "Week of the month" indicates the week in which the productive activity was performed. The table was filled out for each month during data collection.

**Table 5.** Activities of growers based on coffee crop phenology.

| Productive Activities in the Crop | Week of the Month |
|---|---|
| 1. Crop management<br>1.1. Crop management—(Plant nutritional management)<br>1.2. Crop management—(Post-harvesting and reviewing)<br>1.3. Crop management—(Weeds Control) | <br>Week 1<br>Week 2<br>Week 3 |
| 2. Pesticides Control<br>2.1. Fungicides use<br>2.2. Biopesticides use | <br>Week 2<br>Week 3 |
| 3. Polyculture | |
| 4. Ancestral knowledge<br>4.1. Moon use<br>4.2. Bioindicators use | <br>Week 1<br>Week 3 |
| 5. Crop phenology<br>5.1. Germination<br>5.2. Leaf development on main stem<br>5.3. Appearance of the floral organ<br>5.4. Flowering<br>5.5. Grain formation and maturation | <br><br><br><br>Weeks 1, 2, 3<br> |
| 6. Zoca (Method used to renovate coffee plantations. It consists of cutting the stem of the tree about 30 cm from the ground to renew the coffee tree, combat the coffee berry borer bio-aggressor, and improve the yield of the crop) | |
| 7. Productive arrangement<br>7.1. Under free shadow exposure<br>7.2. Under shadow conditions<br>7.3. Under free sun exposure | <br>Week 1<br>Week 2<br> |
| 8. Intercropping<br>8.1. Crop rotation | <br>Week 3, 4 |

In the qualitative dataset, Table 5 outlines the productive activities or tasks of the coffee grower. The following section provides a more detailed description of these activities.

- Crop management: This is the set of agricultural practices carried out to improve the growth and development of crops and the control of bio-aggressors.
- Biopesticides control: Outbreaks of pests and diseases in coffee crops are mainly triggered by different factors, such as the variation in climatic conditions and the inadequate use of fertilizers or biopesticides (fungicides or biopesticides), which creates an imbalance in the beneficial microorganisms of the crop [34]. The poor management of biopesticides on coffee crops contributes to the development of pathogenic fungi, bacteria, or viruses that impede the plant's normal development [35]. Due to this, using biopesticides in crops is considered an essential activity by coffee growers.
- Polyculture: This is a productive agricultural practice where more than one crop is planted on the same land, in which coffee plants are accompanied by crops such as tomato, banana, and avocado, among others. This productive activity aims to identify other crops' influence on coffee plantations.
- Ancestral knowledge: This corresponds to farmer wisdom applied to coffee crops for developing and protecting coffee crops. Table 6 presents the factors considered as ancestral knowledge in coffee crop practices. The column "Week of the month" indicates the week of a month in which the Ancestral Knowledge event occurred. The table was filled out for each month during data collection. For coffee growers, using empirical knowledge to treat crops is even more significant than using information

technologies, since it implies a cost that the small coffee grower cannot afford. Within the practical knowledge they inherited from their ancestors, they expressed that using the moon is essential in their crops, e.g., the best lunar phase to harvest coffee beans to produce seeds is the waning quarter towards the new moon because the fruits have already undergone the best physiological maturation. In case that they are dried and stored, they will resist deterioration. Practices such as these express that, although it has no scientific dissemination, this empirical knowledge has worked well in their crops, not only in the stages of crop development but also in the process of pest and disease control.

**Table 6.** Ancestral knowledge considered in the survey in coffee crops.

| Ancestral Knowledge | Week of the Month |
| :---: | :---: |
| 1. Moon phase 1.1 First Quarter 1.2. Full Moon 1.3. Last Quarter 1.4. Waning crescent 1.5. New Moon | Week 3 |
| 2. Bioindicators use (viewing and analyzing animals' behavior on the crops) | |
| 3. Other ancestral knowledge | |

Coffee growers often perform productive tasks, such as sowing or "zoca", based on lunar phases. Sowing coffee during the waning moon is generally recommended due to the quick growth of new plantations. Monitoring using bio-indicators is crucial as this can detect possible climatic changes and bio-aggressors on the coffee crop based on animal behaviors. Using bio-indicators requires extensive empirical knowledge from coffee growers, who can determine the causal correlation of bio-aggressors on coffee trees based on cultivation events. For instance,

- The presence of the coffee mealybug pest is proportionally incremental to the presence of ants in the crop, presenting symbiosis between the two species, since when the coffee mealybug, consumes the sap with the nutrients of the plant and excretes what is consumed, components are formed that serve as food for the ants. For this reason, if the crop's ant population increases, the coffee mealybug pest's presence can be inferred.
- The presence of birds indicates the possible presence of mites on the crop since some mites are a bird food source.
- From the overflight of the cicada birds, it is possible to infer possible climate changes and rainfall generation.

The inference of possible rainfall or triggering of bio-aggressors is associated with the analysis of the behavior of an animal or an event occurring in the crop:

- Crop phenology: This is essential for the proper planning and management of practices such as fertilization, disease, pest, and weed control, among others. Crop phenology is crucial for identifying coffee crop phases susceptible to pests and diseases [24].
- Zoca: This corresponds to renewing the branches and increasing production by removing an old part of the trunk or branches of a coffee plant [36].
- Productive arrangement: This explains the crop properties related to the slope of the land where the coffee is cultivated (Flat, Hillside, Other). Other properties are coffee in free exposure, under shade (provided by other crops such as banana tree), the stage of cultivation according to the phenology, and the variety of coffee harvested.

### 3.5.3. External Conditions

To identify the causes of pests and diseases in coffee crops, it is crucial to consider external conditions, also known as landscape conditions. We focus on two external factors: animals and water sources near the crops (as shown in Table 7). The presence of water sources can increase humidity levels, which, in turn, can trigger bio-aggressor outbreaks.

**Table 7.** External conditions considered in coffee crops.

| External Conditions to the Crop | Answer |
| --- | --- |
| Water sources near to farm | Yes |
| Other type of crops next to coffee crop | |
| Rivers flow close to the farm | No |
| Mammals or birds living near the coffee crop | Yes |
| Other external conditions to the crop that could have an influence | |

### 3.5.4. Bio-Aggressors

Coffee growers have identified diseases such as berry blotch and brown-eye spots in the bio-aggressors category. Only the CBB pest was detected in the pests category. Weeds were classified into noble and aggressive categories, as seen in Table 8. The column "Week of the month" indicates the week of a month in which the bio-aggressor event occurred. The table was filled out for each month during data collection.

**Table 8.** bio-aggressors in coffee crops.

| Bio-Aggressor | Type | Week |
| --- | --- | --- |
| Disease | Berry blotch | Week 1 |
| Disease | Brown-eye spot | Week 4 |
| Pest | Coffee Berry Borer | |
| Pest | Mites | |
| Pest | Coffee mealybug | |
| Pest | Coffee leaf miner | |
| Weeds | Noble Weed | Weeks 1–4 |
| Weeds | Aggressive Weed | Week 3 |

### 3.6. Data Analysis

This subsection outlines the analysis from statistical and data mining approaches applied to the qualitative dataset. The dataset underwent processing using technologies like Python (version 3.8.12) and R (version 4.0.2). Data pre-processing included procedures for cleaning and enhancing the dataset. This data processing organized qualitative data systematically to facilitate subsequent processing and the application of diverse statistical approaches like the following:

- Hierarchical clustering: Hierarchical clustering with a dendrogram [37] illustrates the arrangement of the clusters produced by the corresponding analyses. Group the data set variables and indicate the variables with greater similarity based on the distance. Moreover, it indicated similarities between variables where we related productive work, weather conditions, and other external conditions of the crop to the triggering of bio-aggressors. However, it was unclear and easy to visualize particular information about the cluster with bio-aggressors, and the visualization was not the best way to share the results.
- Wrapper selection method: The wrapper selection method makes effective feature selection without the problem of having to process vast amounts of data. This method is a novel innovation because it improves classification performance with lower computational cost. Moreover, the Wrapper feature selection method, based on the

S-C4.5-SMOTE sampling method, improves the feature selection and the classification performance of the Bagging C4.5 algorithm [38].

- Decision trees C4.5 algorithm: The C4.5 decision tree algorithm was applied using the results of the wrapper selection method, where 178 instances of the dataset were analyzed, and pests and diseases were present in coffee crops.

These approaches aimed to generate predictions and uncover relationships between variables, including their role in triggering bio-aggressors. Such triggering mechanisms were identified based on behavioral patterns found within the dataset.

Figure 9 illustrates the monthly occurrences of diseases and pests. Within the dataset, 254 instances lack bio-aggressor information, but 128 instances (29.6%) indicate the presence of CBB, 44 instances (10.2%) correspond to the brown-eye spot, and a smaller fraction of the dataset, consisting of 4 instances (0.9%) and 2 instances (0.46%), respectively, reveal occurrences of berry blotch and aggressive weeds.

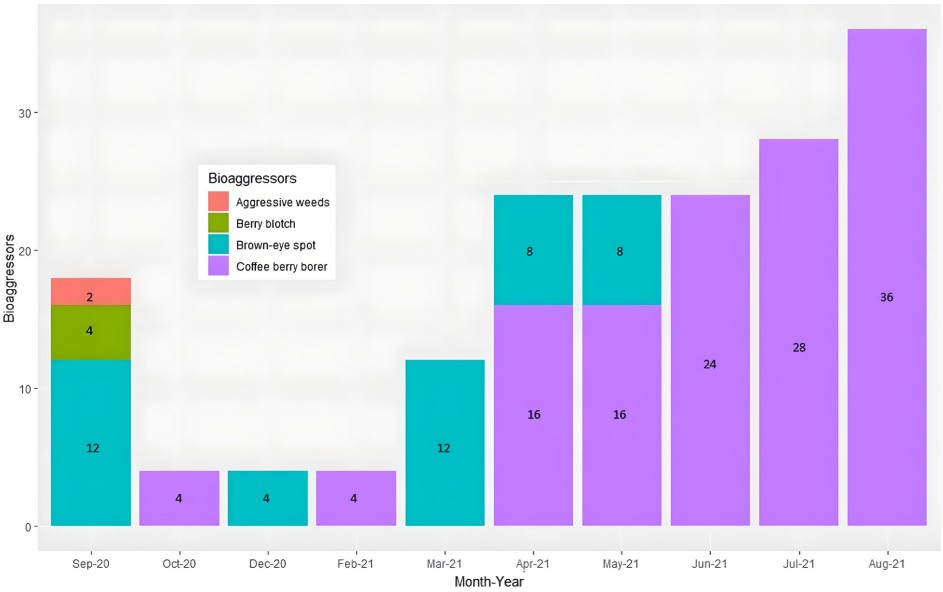

**Figure 9.** Bio-aggressors presented by month.

Figure 9 illustrates the occurrence of bio-aggressors by month. Notably, no bio-aggressors were observed in November 2020 and January 2021. However, in September and December 2020, there were 28 recorded cases of brown-eye spots. This affliction was also observed from March to May 2021. Conversely, the CBB appeared incrementally from April to August 2021. The most significant concurrent bio-aggressor occurrence transpired in September, encompassing three bio-aggressors: brown-eye spot, berry blotch, and aggressive weeds.

### 3.7. Weather Conditions

Figure 10 shows the association between weather conditions and the bio-aggressors during the year.

Aggressive weeds tend to appear in September due to cold temperatures and rainfall. Berry blotch is also associated with cold temperatures, hail, and windstorms. Prolonged cold weather can cause fruit to freeze and burn. Brown-eye spot is most common in September, December, March, April, and May and is linked to cold temperatures and constant rainfall. CBB, on the other hand, occurs during dry periods in October, February, June, July, and August. CBB is more likely to occur in the months following harvest when fruits drop and dry conditions create ideal conditions for CBB outbreaks.

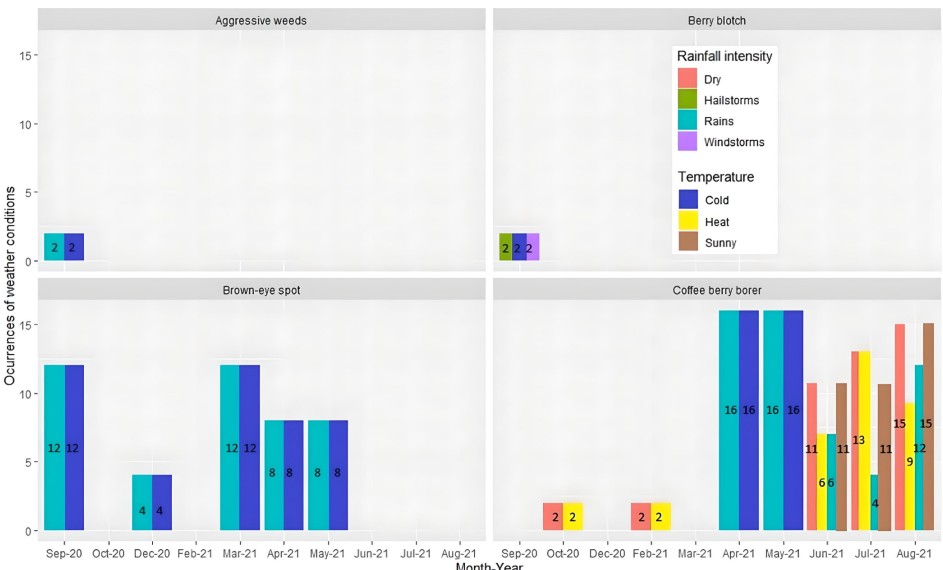

**Figure 10.** Weather conditions (temperature and rainfall intensity) and bio-aggressor occurrences from September 2020 to August 2021.

*3.8. Productive Activities*

3.8.1. Coffee Variety and Bio-Aggressors

Figure 11 illustrates the bio-aggressors' distribution across coffee varieties. Caturra coffee plants exhibited the highest susceptibility to bio-aggressors, with 76 CBB incidents from October 2020 to February 2021 and from April to August 2021. Additionally, 20 instances of brown-eye spot were reported in September and December 2020, as well as in March and April 2021. On the other hand, Tabi coffee plants documented 32 cases of CBB infestation during April, June, July, and August 2021, along with 16 occurrences of brown-eye spot in September 2020 and in March and May 2021.

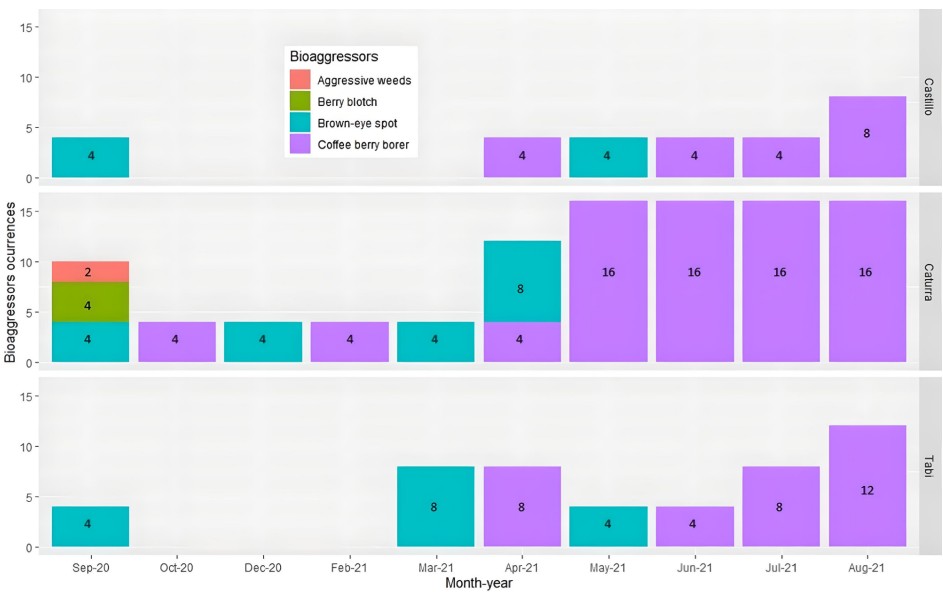

**Figure 11.** bio-aggressors presented by crop variety considering the months of the year.

3.8.2. Coffee Phenology and Bio-Aggressors

Figure 12 shows the occurrences of the bio-aggressors by crop phenology phases (flowering, fruit ripening, harvest, and post-harvest).

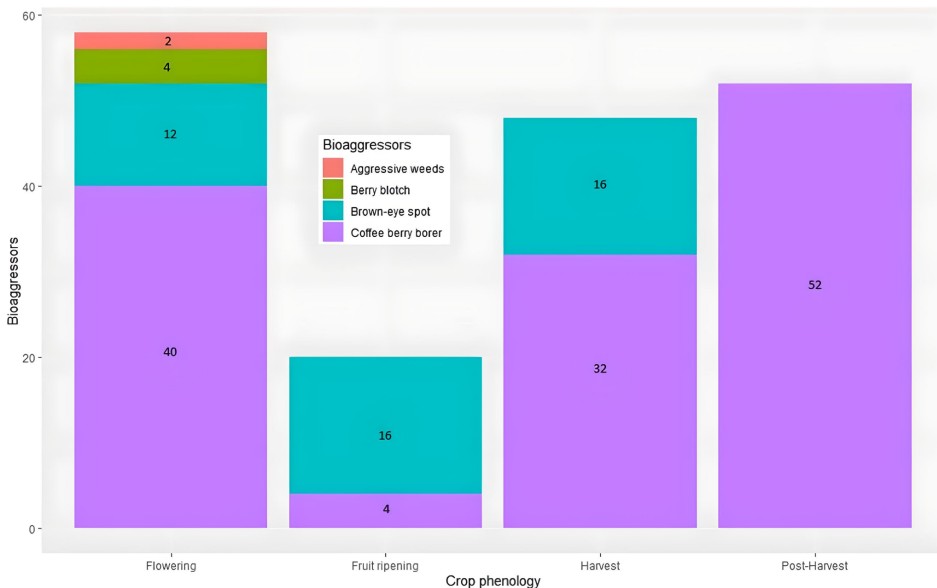

**Figure 12.** Bio-aggressors triggered by phenology.

The brown-eye spot is reported during the flowering, fruit ripening, and harvest stages. Moreover, the presence of aggressive weeds is more frequent during the flowering stage of the crop. On the other hand, CBB occurs in all four stages (flowering, fruit ripening, harvest, and post-harvest). However, the most significant number of incidences are detailed in the post-harvest stage and are correlated with the fall of coffee plants to the ground, which, together with climatic conditions such as drought or heat, trigger CBB.

### 3.8.3. Productive Activities and Bio-Aggressors

The chart shown in Figure 13 displays the bio-aggressors discovered during the farming operations of each farm. It was observed that, in farms 5, 7, and 8, outbreaks of CBB happened concurrently with the activities of "harvesting and reworking". The presence of the bio-aggressor during the post-harvest phenology stage (as seen in Figure 12) suggests that this caused the outbreak. To address this issue, harvesting, and reworking were deemed necessary to control the CBB.

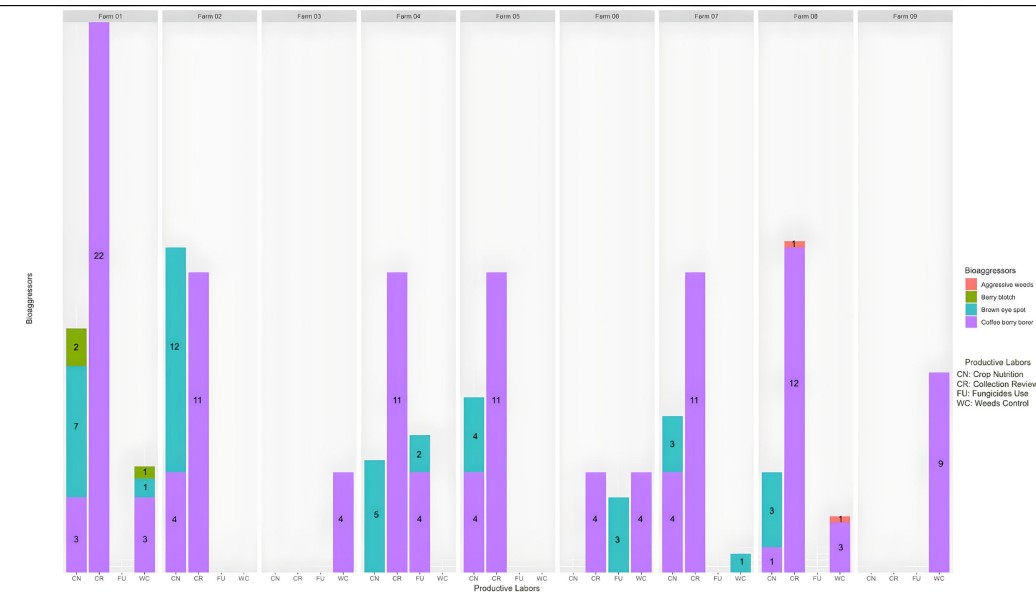

**Figure 13.** Bio-aggressors triggered by productive activities.

Similarly, brown-eye spot presented in farms 1, 2, 4, 5, 7, and 8 when coffee growers applied the productive activities of "crop nutrition". The application of crop nutrition allows for controlling the bio-aggressor by balancing the beneficial micronutrients for the coffee plant.

Finally, the farms most influenced by bio-aggressors were 1, 2, 4, 5, 7, and 8. In contrast, farm 3 presented fewer bio-aggressors (four cases of CBB), while the coffee growers applied weed control and collection review to reduce the pest.

### 3.9. Application of Multiple Correspondence Analysis

MCA, or multiple correspondence analysis, analyzes categorical data to uncover and visualize internal structures within a dataset [39]. MCA identifies relationships between dimensions and variables by plotting them on a graph. When dimensions and variables are close together on the graph, this indicates a strong relationship. Moreover, MCA generates a plane that shows the distance between the dimensions and the origin, revealing that the farther away the categories are from the origin, the stronger the association between the dimensions.

The decision to present the results using the statistical approach, multiple correspondence analysis (MCA), was taken after analyzing the dataset with different statistical and data mining approaches, such as hierarchical clustering with a dendrogram, the wrapper selection method, and a decision tree with C4.5 algorithm. Using statistical and data mining approaches to discover and analyze the qualitative dataset allowed us to see associations and behavioral patterns in the data that we would not have discovered otherwise. Therefore, it was determined that presenting the results under the MCA method would allow for a better visualization of the variables in the dataset. Additionally, we took advantage of the fact that MCA is a method of analyzing categorical data to uncover and visualize internal structures within a dataset.

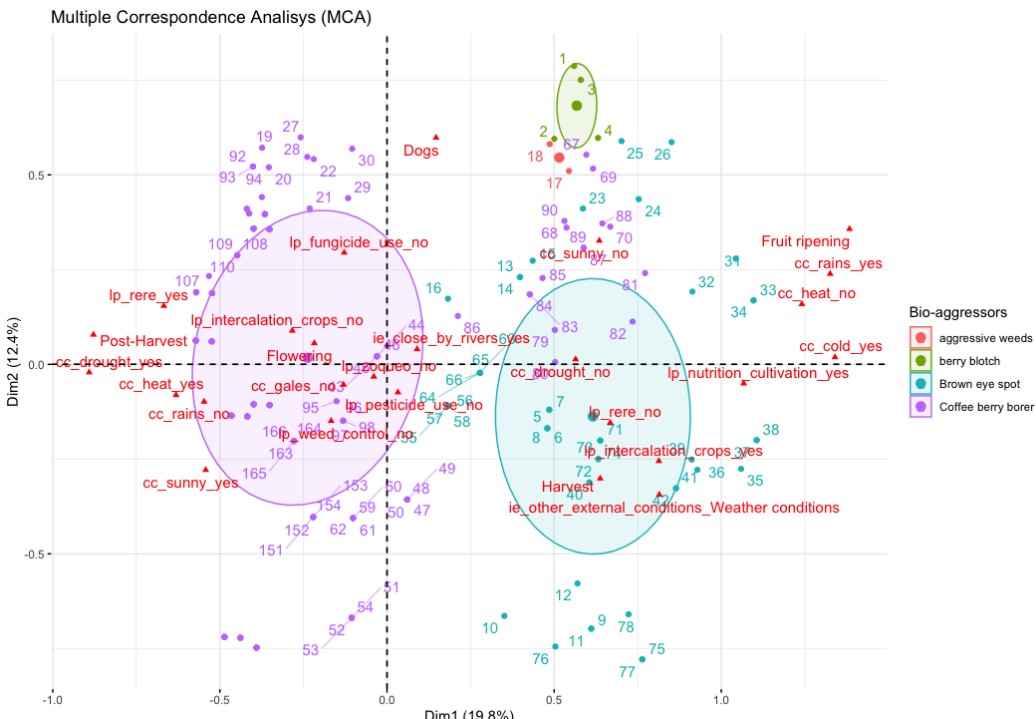

**Figure 14.** MCA overlap map concerning the occurrence of bio-aggressors in coffee.

The MCA plane in Figure 14 shows the proximity relationship between the bio-aggressor known as brown eye spot and several nearby factors, including cold conditions, rain, absence of heat, and presence of clouds. The presence of this bio-aggressor occurs under humid conditions. Additionally, it can be analyzed that it occurs during the

fruit ripening stage. Furthermore, it is observed that the coffee grower regularly performs crop nutrition under those humid conditions to counteract the nutritional imbalance that humidity could cause and avoid making the plants more susceptible to the disease. For this reason, nutrition is applied to the crop to generate plant resistance to damage from cold or humidity and brown eye spots. As for the appearance of the CBB, according to the map analysis (Figure 14), it is related to the incidence of drought, heat, or sunny environments, especially in flowering and post-harvest stages. Moreover, CBB occurrence is related to productive labor such as collecting and reviewing (in Spanish(Recolecta y Recolección—ReRe)), inferring that there is a relationship between that productive work in the post-harvesting stage because that labor helps to reduce the birth and propagation of CBB.

## 4. Conclusions

The knowledge passed down through generations of coffee farmers is an invaluable legacy representing a continuous stream of wisdom [40]. In Colombia, this knowledge is deeply ingrained in the fabric of their culture and is tied to their spirituality, identity, practices, economy, and way of life.

Within this context, we propose a qualitative dataset that will be a groundbreaking effort to capture the profound knowledge of Colombian coffee farmers regarding cultivation techniques, weather patterns, and the interaction of pests with the crops. We aim to understand complex concepts, experiences, and opinions that agricultural sensors cannot measure.

In addition, we uncovered behavioral patterns in the data that suggest various causes of bio-aggressors in coffee crops. This provides an opportunity to implement corrective measures in the productive tasks related to the potential causes of bio-aggressors in coffee.

Despite having only 432 records, the dataset does not lose weight, since it can be applied to any exploratory or predictive analysis based on the dataset. Therefore, it is concluded that not all datasets with too many variables and records allow any analysis and even establish patterns of the internal behavior of the data in order to contribute to the main focus, which is the possibility of establishing relationships for triggering bio-aggressors in coffee crops.

Finally, to detect qualitative information on bio-aggressor control in coffee crops based on ancestral knowledge, the dataset comprehensively examines diverse criteria encompassing climatic conditions, productive tasks, and external influences on the crop. These criteria form the bedrock of a coffee grower's daily routine. Through scrutinizing these elements, invaluable insights can be derived, facilitating informed recommendations for optimal crop management and mitigating detrimental practices. This, in turn, is a robust measure against bio-aggressors' proliferation within the coffee plantation.

## 5. Future Work

In the realm of future endeavors, we envisage the application of association rules algorithms to our qualitative dataset. The realm of association rule mining, renowned for its adeptness in unveiling frequent patterns, correlations, associations, and even causal structures within qualitative datasets, holds significant promise for our research. This approach can unearth hidden insights and relationships that might be instrumental in comprehending the intricate dynamics inherent in our data.

Moreover, delving into machine learning techniques presents an avenue for enhancing the predictive capabilities of our dataset. These methods could potentially distill the wealth of qualitative information in the data into predictive models, aiding in forecasting possible occurrences of bio-aggressors and guiding more effective agricultural decisions [41].

**Author Contributions:** Conceptualization, J.F.V.-M., D.G., M.S.-M., C.F., J.C.C. and D.C.C.; methodology, J.F.V.-M., D.G., M.S.-M., C.F. and D.C.C.; validation, J.F.V.-M., D.G., M.S.-M., C.F., J.C.C. and D.C.C.; investigation, J.F.V.-M., D.G., M.S.-M., C.F., J.C.C. and D.C.C.; resources, M.S.-M.; data collection, J.F.V.-M. and M.S.-M.; writing—original draft preparation, J.F.V.-M., D.G., C.F. and D.C.C.; writing—review and editing, J.F.V.-M., D.G., D.C.C. and C.F.; visualization, J.F.V.-M., C.F. and D.C.C.; supervision, D.C.C. and C.F. All authors have read and agreed to the published version of the manuscript.

**Funding:** Author Juan Felipe Valencia-Mosquera was financed by Call 823—High-level training human capital for the regions of Cauca. We recognize the Ministry of Science, Technology and Innovation (MinCiencias). Furthermore, this work is part of the research project titled "Enhancement of Technological Prototypes in the Precommercial Stage Derived from R&D Results for the increase of the agricultural sector in the department of Cauca". This research project is funded by the General Royalties System (SGR), code BPIN 2020000100098.

**Institutional Review Board Statement:** Not applicable.

**Informed Consent Statement:** Not applicable.

**Data Availability Statement:** The data presented in this study are openly available in Zenodo at https://doi.org/10.5281/zenodo.8275090 (accessed on 28 September 2023).

**Acknowledgments:** We would like to thank the Association of Agricultural Producers (ASPROACA) for their participation in the data collection and for providing the interviews and feedback of this work.

**Conflicts of Interest:** The authors declare no conflict of interest.

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
