# Peer review of "A Qualitative Dataset for Coffee Bio-Aggressors Detection Based on the Ancestral Knowledge of the Cauca Coffee Farmers in Colombia"

_data, 2023_

Round 1

Reviewer 1 Report

 The title of the article pertains to coffee pest detection. But the data presented in Figures 4-8 focus on bio-aggressors, which encompass pests, diseases, and weeds. Please to ensure consistency between the title and the content presented in the paper.

The provided dataset link in the paper does not lead to a downloadable dataset. Additionally, the article does not display the recorded data for reference. Please provide a link to downloadable datasets and display the recorded data in the paper.

The article outlines the impact of each variable on coffee production. However, there is a lack of specific quantification for the quantity of each variable within the dataset. Please consider including a summary of the data quantities for each variable.

The constructed dataset in the article encompasses multiple variables, but the statistical analysis is limited to bio-aggressors and weather data. It would be valuable to extend the analysis to include other relevant variables within the dataset. This would provide a more comprehensive understanding of the factors influencing coffee production. Consider expand the scope of data analysis and statistics.

The utilization of Multiple Correspondence Analysis (MCA) is noted in the methodology. However, the rationale behind selecting this method is not explained. Please consider including a brief explanation of why MCA was chosen.

Minor editing of English required

Reviewer 2 Report

Dear colleagues,

I believe your dataset, text and ideas may be useful for wide audience, but I am sure that the text may be improved. My guess is that the authors declare importance of traditional knowledge for farming, however, actually they discuss the role of this knowledge very briefly. Besides, there is some scientific problems, because the role of the lunar stages relative to activity of many species is under a question (e.g., how they can affect development of plants?).

There are some other problems as well.

The dataset per se was removed from ZENODO. As a result, reviewers can't to check it.

line 16 — Coffea arabica L. — please, use italics for the Latin names of species but authors should be in roman.

lines 25–26  — above 1,400 m — do you mean between 1,400 and 1,500 m?

Figure 1 — please, add scales

line 89 — 'climatic conditions' duplicate

line 91 — instances — records (?)

lines 100–101 — foundations > bases  

Table 3 and Figure 3

level curved slope — terraced slope (?)

temperature  — 'sunny' is not characteristic of temperatures

'Zoqueo' — please, explain this term

'as moon use' — do you mean lunar stages? or something else?

line 117 agronomic areas > agronomic characteristics (?)

line 118–125 — you repeat in the text some information described on Figure 3 — please, avoid such situations!

lines 126–127 and 134–135 and so on — almost the same! again and again you repeat almost the same words!

Tables 4–6 — what means 'week of the month'? e.g., in the Table 4, 'Heat' occurs during the first week of each month?

line 182 — spiders > mites

line 200 and so on, Table 7 — 'animals' — what do you mean? Mammals? Vertebrates? Ants and mites are animals as well.

line 219 — you have about 400–500 records, that means there is no sense in the third and fourth digits after points.

line 229 — please, use 'association' instead of 'correlation'

Figure 9 — please, to magnify this picture     

No special comments

Reviewer 3 Report

The manuscript describes a dataset compiling tacit knowledge from coffee farmers in the region of Cauca, Colombia. The dataset is interesting, but the manuscript has few weaknesses that need to be addressed, as detailed below.
- The survey design needs to be explained in much more detail. The authors claim that a comprehensive literature review was performed, but not of such literature seems to be mentioned in the manuscript. It is also not clear how was the process of including and excluding variables from the survey.
- Only nine coffee farmers took part on the research. It is indeed rather difficult to involve farmer in this type of research, but such a small number will certainly limit the representativity of the dataset. Those limitations should be recognized and thoroughly discussed in the article. Also, while age and gender information were provided, other important information like level of formal education and level of access to technology were not provided. These are important for proper characterization of the sample.
- The authors should explore in more detail the potential benefits that can potentially arise from the proposed dataset. The authors mention some generic benefits, but the actual practical value of the dataset is not made clear.
- There are several uncommon terms that have not been properly defined, like “zoqueo” and “cosmovision”.
- Most figures are too small and fonts are mostly illegible.

Language needs considerable improvement, as there are several grammatical errors and strangely constructed sentences.

Round 2

Reviewer 2 Report

Dear authors,

I suggest some ideas to improve the last version of your manuscript:

(1) Coffea arabica L. — when you use the formal name of some species, you should differ the species name per se and the name of the author(s). Please, use italics for the Latin names of species but authors should be in roman.

(2) In many cases, you try to use the word sunny as the characteristics of temperatures. Actually 'sunny' is a kind of weather without clouds. In some regions (e.g. Canada and Siberia) the weather may be sunny, but temperatures may be about minus 20 C. Please, remove this word from the characteristics of temperatures.

(3) Tables 4-6 - you should explicitly explain what means 'week of the month'? e.g., in the Table 4, 'Heat' occurs during the first week of each month?

(4) You commonly discuss the  red spiders, but really the red spiders are the family of mites (not spiders). This means when you write about this taxon you can use the 'red spides', but when you consider the mites as the order, you should call them mites.

No comments

Reviewer 3 Report

Most of my suggestions were properly addressed. However, there are two aspects that can still be improved. First, most of the figures are still too small; the fonts, in particular, are too small in most figures, making it very difficult to read the relevent text. Second, there are still some language issues that could be corrected with a careful revision.

There are still some language problems that could be corrected with a careful revision.
